# Habitat Suitability Based Models for Ungulate Roadkill Prognosis

**DOI:** 10.3390/ani10081345

**Published:** 2020-08-04

**Authors:** Linas Balčiauskas, Jack Wierzchowski, Andrius Kučas, Laima Balčiauskienė

**Affiliations:** 1Laboratory of Mammalian Ecology, Nature Research Centre, Akademijos 2, 08412 Vilnius, Lithuania; andrius.kucas@gamtc.lt (A.K.); laima.balciauskiene@gamtc.lt (L.B.); 2Geoprofis Austria, Stubenerstraße 73, 7434 Stuben, Austria; jack.wierzchowski@gmail.com; 3European Commission, Joint Research Centre, Via E. Fermi 2749, I-21027 Ispra, Italy

**Keywords:** roe deer, red deer, wild boar, habitat suitability, transport accidents

## Abstract

**Simple Summary:**

Red deer, roe deer and wild boar movements and crossings of the two highways in Lithuania were modeled. Validity of obtained models was tested by comparing the pathway predictions to the real roadkill and roadkill cluster locations in 2002–2009 (at the time the models were created) and in 2010–2017 (testing the prognostic value of these models). Across both periods and on both highways, the roe deer roadkill locations were significantly closer to the model-predicted pathways than to randomly selected points. The prediction of roadkill locations was also good for wild boar. The roe deer roadkill clusters and multi-species clusters were significantly better represented by the model than by random distribution. Thus, habitat suitability-based models of ungulate movement are recommended as an additional tool for planning wildlife-vehicle collision mitigation measures.

**Abstract:**

Roads do not only have a detrimental effect on nature (fragmenting habitats, isolating populations and threatening biodiversity), but the increasing numbers of wildlife-vehicle collisions are also a direct threat to humans and property. Therefore, mitigation measures should be placed with respect to animal distribution and movements across the roads. We simulated red deer, roe deer and wild boar movements in Lithuania, focusing on the two main highways A1 and A2. Using regional habitat suitability and linkage models, we calculated movement pathways and the most probable crossing zones in 2009. The prognostic value of these models was tested by comparing the pathway predictions to the real roadkill and roadkill cluster locations in 2002–2009 and 2010–2017. Across both periods and on both highways, the roe deer roadkill locations were significantly closer to the model-predicted pathways than to randomly selected points. The prediction of roadkill locations was also good for wild boar. The roe deer roadkill clusters and multi-species clusters were significantly better represented by the model than by random distribution. On both highways, the biggest differences in distance from the predicted locations were near big cities. We recommended wildlife movement models as an additional tool for planning wildlife-vehicle collision mitigation measures and we advise measures for increasing their predicting power.

## 1. Introduction

As part of the transportation infrastructure, roads have well-known detrimental effects on wildlife, including mammals [1,2,3]. Their impacts are well-documented worldwide [4,5,6]. In an increasing number of landscapes, movements of animals, particularly wide-ranging large mammals, come into contact with and require the crossing of roads. As new roads are built and old ones upgraded to accommodate greater traffic demands, the rate of successful animal crossings decreases, becoming in some cases the leading cause of animal mortality [7,8,9]. Such barrier effects contribute to habitat fragmentation, demographic isolation and loss of biodiversity [10,11].

Various EU initiatives such as the COST-341 program (Habitat Fragmentation Due to Linear Transportation Infrastructure) [12], the CEDR initiated Saferoad program in 2013 (https://www.saferoad-cedr.org/en/saferoad.htm) and the Transnational Research Programme “Conflicts along the Road: Invasive Species and Biodiversity” in 2016 (https://cedrprogress.eu/project-description/) have heightened the urgency for sustainable transport systems to incorporate mitigation structures into transportation planning schemes [13,14,15]. However, as transportation planning was, and is, generally considered a one-dimensional linear zone along roads and highways, engineering and design dimensions have always been the primary concern for planners. However, the ecological effects of roads are many times wider than the road itself and can be immense and pervasive [11]. Because of the broad landscape context of road systems, it is essential to incorporate landscape patterns and processes in the planning and construction process [16].

When used in a geographic information system (GIS) environment, regional or landscape level connectivity models of sufficient resolution can facilitate the identification and delineation of barriers and corridors for animal movement [17,18,19]. This provides for the development of a more integrated land use strategy by taking into account different land management practices and prioritization of habitat conservation concerns [20,21]. The mentioned approach promotes a more integrated, larger scale methodological procedure that contemplates landscape patterns and processes in the planning of transportation infrastructure [4,16].

With increasing availability of digital habitat suitability and biophysical and land-use data, GIS tools and applications are becoming more popular among resource managers and transportation planners for amassing information and modeling impacts and habitat linkages across roads [22,23,24,25]. Wildlife Movement Simulation (WMS) models, being cost effective, are a natural extension of these tools [18,26,27]. In highly fragmented ecosystems, WMS models not only help locate wildlife movement corridors, but also permit their ranking in terms of their ability to sustain wildlife movement. They may also help determine the economics of maintaining such corridors as viable wildlife migration routes. Combined with an analysis of forestry and land use practices, WMS models can help to provide explicit guidelines on how to maintain, enhance or even restore connectivity between isolated wildlife habitats [28,29].

To evaluate movement and roadkills of red deer (*Cervus elaphus*), roe deer (*Capreolus capreolus*) and wild boar (*Sus scrofa*) in Lithuania, focusing on the two main highways, we applied a North American WMS [18]. The aims of the paper were: (1) to present habitat suitability- and linkage-based movement models of three ungulate species, developed in 2009; (2) to test the prognostic value of the models in terms of recommendations for mitigation measures and the placement of wildlife crossing structures. Testing was done by comparing the model’s predictions with the actual locations of roadkills and roadkill clusters of these species in the period either side of the model development (2002–2009 and 2010–2017).

## 2. Materials and Methods

We used the following scheme to model ungulate movements across the highways: (1) habitat ranking for ungulates (based on literature data and expert opinions), (2) assessment of human impact, (3) habitat suitability calculations, (4) modeling of the animal movements.

### 2.1. Study Area

We assessed habitat suitability and simulated ungulate movements along the 311 km A1 highway (part of the European E85) connecting Vilnius with Klaipėda, and the 136 km A2 highway (the E272, the only European highway located entirely within the country) linking Vilnius with Panevėžys (Figure 1). For the study area, we used a 20 km buffer on each side of the highway for *C. elaphus*, an 8 km buffer for *S. scrofa* and a 3 km buffer for *C. capreolus*. Buffer width was chosen according to expert opinion (through a questionnaire distributed among wildlife biologists and forest service experts familiar with the investigated species) and literature data on the species biology (data and references presented in the Appendix A). The buffer areas covered 27.86%, 11.04% and 4.09% of the total country area, respectively. Forest cover within these buffers amounted to 31.67%, 28.56% and 27.32%, respectively, and agricultural areas 59.13%, 60.80% and 61.16%, respectively. Ungulate movements across the highways A1 and A2 were also affected by wildlife fencing, with the length of these fences increasing by 19.34% between 2009 and 2017. Total length of the wildlife fence on the both sides of the highway A1 was 16.34 km in 2002, 168.48 km in 2009, and 245.161 km in 2017; length of fences on A2 was 0.374 km, 231.298 km and 243.813 km, respectively. The locations of the fences and underpasses in these years are shown in Figure 1.

### 2.2. Assessment of the Habitat Suitability

Habitat suitability rankings (Table 1) were based on literature data, summarized in Appendix A, and expert opinions. These rankings reflect the relative importance of various habitat components digitally mapped within the study area. We found little quantitative information about wild boar habitat selection (Appendix A). Hence, in comparison to the other models, which are also somewhat subjective, we recognize a higher degree of uncertainty in the wild boar habitat suitability rankings presented in Table 1. These rankings can be further refined when new, quantitative information on wild boar habitat selection becomes available. Wild boars primarily select deciduous and mixed forests with oak inclusions (Appendix A).

### 2.3. Assessment of the Human Impacts

Human impacts were analyzed with regards of the species’ response to (i) roads and railways and (ii) built-up areas, using 25 literature sources, which are cited in Appendix A.

Human impacts on red deer were assessed using the moving window technique with the circular kernel of a diameter equal to the average distance of red deer daily movement (4 km). We established the mathematical relationship between road density and habitat utilization by fitting a polynomial curve (Figure 2a) to the scatter-plot of road density and habitat utilization values as reported by [30]. In the absence of empirical data, we conservatively assumed that red deer habitat utilization would drop to zero if more than 50% of the area (within the kernel) consisted of human habituation footprints, such as towns or villages (Figure 2b).

We found no published quantitative information regarding roe deer response to human infrastructure. The literature points to the species having a high tolerance of roads and railways, but does not address the question of avoidance of built-up areas (Appendix A). We attempted to address this issue by analyzing available data on roe deer road mortality (data for Lithuania, 2002–2007). The median distance to the nearest town or village from roe deer roadkill locations was calculated, and this median distance was treated as a cut-off point below which roe deer use land only marginally.

As for the assessment of wild boar response to human infrastructure, the literature only points to the species’ low tolerance of roads, railways and built-up areas (Appendix A). Analysis of wild boar roadkill locations (data for Lithuania, 2002–2007) showed them to have occurred 500 m away (median value) from villages and towns, but the low sample size (*n* = 99) and the associated high standard deviation point to a high degree of uncertainty with these results. In the absence of any other quantitative data, we decided to use the better documented roe deer buffer of 210 m to reduce the habitat rank for wild boar by 50% around towns and villages.

### 2.4. Usage of the NDVI Index

For red deer, we used the Normalized Difference Vegetation Index (NDVI, the most well-known and used index to detect live green plant canopies in multispectral remote sensing data), rescaled from 0 to 1, at 0.01 increments, to map potential riparian movement corridors. We defined them as the portions of 25 m wide buffers along major rivers and creeks that exhibited NDVI values equal to or higher than the median value (0.4) characteristic to meadow/shrub/deciduous vegetation complexes—the ecotypes that are known to provide good hiding cover for red deer [31].

Roe deer rely on highly digestible, high protein food sources. The distribution of the species’ “field” ecotype is therefore likely to be a function of crop/pastures distribution [32]. We used the NDVI index (correlated with leaf area index) to map the distribution of high biomass crops within the study area. While crop planting patterns change from year to year, such changes are to a large degree restricted by the distribution of immutable soil types and moisture regimes. Hence, the NDVI index calculated for the August–September period (the date of ETM data used) was likely to be reflective of the year-to-year crop patterns within the study area.

### 2.5. Effective Habitats for the Three Ungulate Species Concerned

We calculated habitat suitability (HSI) using the following equations:Red deer HSI = RED_h_ × R_df_ × B_df_(1)
Roe deer HSI = [(ROE_h_ + NDVI)/2] × B_a_(2)
Wild boar HSI = [(WB_h_ + NDVI)/2] × B_a_(3)
where RED_h_ = red deer habitat ranking layer (calculated from the Table 1 using expert opinions)

ROE_h_ = roe deer habitat ranking layerWB_h_ = wild boar habitat ranking layerR_df_ = road density factor (equation shown in Figure 2a)B_df_ = built-up areas density factor (equation shown in Figure 2b)NDVI = Normalized Difference Vegetation Index rescaled from 0 to 1 at 0.01 incrementsB_a_ = Built-up areas factor (50% reduction of habitat rank within 210 m around built-up areas).

### 2.6. Animal Movement Component of the Model

We expressed wildlife movement as a function of topography, habitat quality and the distribution of natural and infrastructure-related barriers. The movement component was based on the “least-cost” principle, which states that wildlife movement is most likely to follow a path of least resistance. Modeling was presented in detail in [27], thus, here we present only a short summary of this method.

We converted our habitat maps into a friction surfaces using the empirically derived formulas to convert habitat maps to the “resistance” or “friction” grids [27]. Formulas ensured that the generated pathways closely followed high quality habitat patches expressing high theoretical forage potential:when HSI = 0.00–0.40, friction value *y* = 5.12e^−6.9315 × *HSI*^(4)
when HSI = 0.41–0.50, *y* = −2.88 *HSI* + 1.472(5)
when HSI = 0.51–1.00, *y* = 1.024e^−6.9315 × *HSI*^(6)

Friction grids were further modified to include known barriers (Table 2) to wildlife movement. We used 1:10,000 digital topographic maps of the study area to extract human land use features that were deemed uncrossable barriers to ungulate movement, such as contiguous built-up areas or industrial sites. The model was run twice: the first run with the barriers listed in Table 2 (hereafter the “no fencing model”), then the second run with added wildlife fences as an impermeable barriers and wildlife underpasses as possible migration routes (hereafter the “fenced model”).

We modeled likely movement trajectories between “entry” and “exit” locations on the opposite sides of the investigated highways. The points were generated randomly within the high-quality habitat patches (the top 40 percent of habitat). These random locations were further constrained by the user-specified buffer zone around the reference highways: 2–20 km for red deer, 1–3 km for roe deer and 2–8 km for wild boar. These distances are related to minimum areas that could support the long-term existence of the species populations [33,34,35]. The spatial location of these points is provided in Appendix A.

To capture the density of wildlife populations, the number of generated per-patch locations was proportional to their size. The number of used entry and exit locations for highway A1 was 993 and the number of generated movement pathways was 57,640 (for all three species); the respective values for highway A2 were 418 and 23,536. Detailed information is presented in Appendix A.

We used the algorithm that maximized median HSI values summed up along the entire pathway length, while providing for the shortest possible route connecting any given entry and exit points. For any given pair of entry–exit points, there were five model iterations resulting in alternate movement trajectories. The first iteration simulates the least-cost movement pathway with no obstructions imposed. In the second iteration, the first pathway is blocked forcing the creation of a new pathway distinct from the original. In the third iteration, the first two pathways are blocked and an alternative route is taken, and so on.

For each pathway, a number of user-specified attributes were analyzed: (1) the pathway distance to built-up areas (footprint of towns and villages); (2) the pathway distance to hiding cover (forests, shrubby vegetation, etc.); (3) habitat quality traversed by the pathway; (4) overall pathway length; (5) the ratio of the shortest distance between the given entry-exit locations to the given pathway length: the more “convoluted” (i.e., departing from a straight line) the pathway, the smaller the ratio; (6) the ratio of the shortest distance between any entry/exit locations within the study area to the pathway length, this ratio relates all pathway lengths to the shortest possible distance between high quality habitat patches found on opposite sides of a given highway; (7) the number of hardtop roads/railway tracks crossed by a given pathway; (8) the proportion of pathway length in hiding cover areas.

To generate weights for the pathway attributes, Saaty’s index was used. It produces the weights by means of the principal eigenvector of the pairwise comparison matrix [36]. This procedure generated an internally consistent set of weights and produced an index (consistency ratio) that estimated the probability that the weights were not assigned randomly (Table 3).

Finally, each highway was divided into 1 km segments (according to kilometer markers on the highways, which are stable and more easily recognized than coordinates). For each segment, the number of pathways crossing it and their cumulative statistics (average of potential and realized habitat quality, distance to human land use, pathway complexity and proportion of pathway within forest cover) were calculated. All segments with the above-average crossing frequency (Table 4) were identified as potential conflict areas. It is important to bear in mind that the described method of mapping wildlife movement does not seek to mimic individual wildlife routes across a landscape. The intent of this model is to describe statistically the most efficient potential migration routes that are based on well-established principles of habitat utilization.

### 2.7. Model Testing

We tested the predictive powers of our models by comparing the locations of actual red deer-vehicle collision (REDVC), roe deer-vehicle collision (ROEVC) and wild boar-vehicle collision (WBVC) sites as well as roe deer, wild boar and multispecies clusters in 2002–2009 to identified locations of high frequency crossing zones, which were defined as highway segments that registered above-median frequencies of crossings. As reference, the central point of each cluster (Appendix A) was used. We generated a random set of locations that equaled the number of sites and clusters for each species, using https://www.random.org/integers/, starting from 10th km of the A1 and A2 (to exclude any influence of Vilnius as a city barrier), ending at 290 km of A1 to exclude the influence of Klaipėda city barrier and at 128 km on the A2 to exclude the influence of Panevėžys city barrier. Distances from both sets of points were calculated to the high frequency crossing zones as defined by our models. Our null-hypothesis (H_0_) stated that the actual locations of REDVC, ROEVC and WBVC roadkill locations and cluster sites would be closer to the simulated red deer-, roe deer- and wild boar-road crossing zones than would random sample locations. The alternative hypothesis (H_a_) states that there would be no difference between above mentioned distances, i.e., random locations are as good in describing roadkill and cluster locations as locations predicted by models.

To evaluate the prognostic capability of the model, we compared the actual ROEVC and WBVC sites as well as roe deer, wild boar and multispecies clusters in 2010–2017 with the same locations of the identified high frequency crossing zones (Appendix A). As clusters of REDVC were absent, we compared only locations of the roadkills. The presence of wildlife fences built after 2010 was taken into account.

Clusters were defined as short and significant road sections where concentrated roadkills of the analyzed species occurred. REDVC, ROEVC and WBVC data clustering was done using KDE+ software [37,38]. Finally, using standard Spatial Analysis tools in a GIS environment, we located the REDVC, ROEVC, WBVC and multispecies roadkill clusters and checked if they were within the fenced road sections.

We used a Wilcoxon matched pair test with continuity correction; the distribution of distances were evaluated with chi-square statistics. Cluster lengths and cluster strengths (presented as average ± SD and 95% confidence intervals) were compared using ANOVA and post-hoc test for unequal sample size [39]. Calculations were done in Statistica for Windovs, ver. 6 (StatSoft, Inc., Tulsa, OK, USA).

Null-hypothesis and alternative hypothesis were formulated according [40]. Effect sizes were tested using Excel program ES_calculator (http://mason.gmu.edu/~dwilsonb/ma.html, accessed 26 July 2020), recommended by Nakagawa and Cuthill [41]. We used two dimensionless effect size statistics: d statistics (standardized mean difference) and r statistics (correlation coefficient), and their respective values as benchmarks, which were considered to be ‘small,’ ‘medium’ and ‘large’ effects (r = 0.1, 0.3, 0.5 and d = 0.2, 0.5, 0.8, respectively) [41].

## 3. Results

### 3.1. Habitat Suitability for the Three Ungulate Species in the Buffer Zone

In comparison to the other two species, we found most suitable habitat for the red deer to be quite limited in the buffer zone (Figure 3), with above average habitat patches also being fragmented (Figure 3a). The ubiquity and ecological plasticity of the roe deer was well reflected in the habitat suitability rankings (Figure 3b): above average habitat patches were well connected and abundant. Even recognizing the higher (than in the other two species) degree of uncertainty in the wild boar habitat suitability rankings, we still found abundant above-average habitat that was well-connected. Based on the best habitat patches, the spatial distribution and location of entry–exit points for movement simulation were defined (Appendix A).

### 3.2. Ungulate Animal Movements across Highways A1 and A1 in 2002–2007, as Found in the Habitat Models

To present locations with above average crossing intensity for red deer, roe deer and wild boar, we used four zones in the highway A1 and two zones in the highway A2 for average values (Table 4). Subdivision was based on the location of large populated places, changes in highway alignment and a general pattern of simulated movement.

Simulated red deer movement within all zones on both the A1 and A2 highways was characterized by a high and very consistent average overall pathways rank, with the highest values in the zones A1-4 and A2-2 (Table 4). Roe deer movement was characterized by a lower and more variable average overall pathways rank than that of red deer, with highest values again in the zones A1-4 and A2-2. Simulated wild boar movement within zone A1-1 was characterized by an average overall pathways rank, while by a below average and highly variable rank within zone A1-2, low overall rank in zone A1-3, consistently high rank in zone A1-4, average and consistent rank in zone A2-1 and by the highest and variable overall pathways rank in zone A2-2 (Table 4). Detailed data are presented in the Appendix A.

The frequency of highway crossings by red deer, roe deer and wild boar is presented in Figure 4. Here maps do not reflect the position of all modeled pathways, which were analyzed at a much finer scale. Details of the pathways show the various possibilities of the application of wildlife movement models. First of all, the pronounced differences in the pathway trajectories and densities characterized the influence of the WVC mitigation measures (wildlife fences and wildlife underpasses), as well as the limited permeability of the built-up structures. Secondly, the pathway density and configuration depended on the habitat quality and fragmentation.

For example, in the zone A1-4, the highway was extensively fenced (244–296 km) already in 2007. The single wildlife underpass was positioned at 274 km. In the entire mentioned area, the highway on the both sides is surrounded by above average red deer habitat (Figure 3). Simulated pathways of the red deer were distorted by the fence and depended on the placement of the wildlife underpass (Figure 5). The movement model showed that without wildlife fencing nearly all of the 240–280 km section of the highway A1 part would be used intensively by red deer (Figure 5a). Thus, the placement of the existing wildlife underpass at 279 km is most apt in terms of position for a crossing zone and has shown significant movement through this area. However, the models also clearly indicated that a single underpass within a 40-km section of the highway is not enough to allow relatively unhindered cross highway movement of red deer. Apart from a huge spiking of crossing frequencies at 274 km (Figure 5a), the fenced model registered a “spillover effect” where simulated movement often paralleled the fenced highway and crossed it at the locations (244 km in particular) where the fencing ended (Figure 5b). In the 255–270 km section, the fenced model showed red deer movement pathways being parallel to the highway (Figure 5c), and concentrated at the wildlife underpass (Figure 5d).

Some other examples of the detailed pathway locations are presented in Appendix A. The no fencing model for the wild boar on the 15–33 km stretch of highway A2 showed that segments at 22, 32 and 33 km appear to have the best conditions for supporting cross-highway movement (Appendix Aa). However, the fenced model showed that most movement channeled through 22 km (wildlife underpass) and 28 km (the beginning of the fence). Given the habitat rank at 22 km is one of the highest in the area, this is the most apt location for the underpass.

Zone A2-2 accounted for 71% of all simulated red deer movement, as it is an area of a less fragmented red deer habitat of an overall higher quality (see Figure 3). The resulting more diffused movement pattern produced a higher (than in zone A2-1) number of high frequency crossing zones (Figure 4a). The crossing zone at 115–116 km exhibited relatively high pathway ranks, while the spill-over effect was best expressed at 114, 118 and 128 km (Appendix Ab). This highway segment would be optimal for the placement of a new under or overpass capable of facilitating cross-highway red deer movement.

### 3.3. Roadkills and Roadkill Clusters in 2002–2009 and 2010–2017: Was the Model Predictable?

On the highways A1 and A2, we analyzed 26 REDVC, 765 ROEVC and 185 WBVC. For the roe deer and wild boar, 103 and 12 clusters were found, respectively, while accidents with red deer were not clustered. Number of multispecies clusters was 143 (Table 5). Compared to 2002–2009, the number of ROEVC on the A1 increased by 210% in 2010–2017, while that of WBVC by 129%. On the A2, however, these numbers decreased by 8% and 205%, respectively. On the A1, the number of ROEVC clusters nearly quadrupled in 2010–2017. The average cluster length and cluster strength for ROEVC and WBVC did not change significantly from 2002–2009 to 2010–2017, neither on the A1 nor on the A2. Respective confidence intervals of these parameters between compared periods are overlapping (Table 5); however, all results are significant, as CI did not include zero values.

Effect size (ES) analysis showed that cluster length and strength for the wild boar decreased in both A1 (ES small) and A2 (ES large). For the roe deer, cluster length and strength increased in A1 (ES small and medium, respectively), cluster length did not change and cluster strength decreased (ES small) in A2. Multispecies cluster length and strength increased on both A1 and A2 (ES small or not significant). Detailed data on the roadkill clusters are presented in Appendix A, detailed data on the effect size in Appendix A.

The spatial representation of multi-species crossing zones (Figure 6a), REDVC, ROEVC and WBVC roadkill sites (Figure 6bc), as well as ROEVC and WBVC clusters (Appendix A), was compared to the simulated movements of three ungulate species across the highways A1 and A2 in Lithuania as of 2007 (Figure 4). Roadkill locations were significantly closer to model-predicted pathways than random locations in 2002–2009, both on the A1 and A2 for roe deer (1.22 ± 0.21 vs 1.89 ± 0.29 km, Wilcoxon z = 2.28, *p* < 0.025 and 0.81 ± 0.08 vs 1.44 ± 0.12 km, z = 4.94, *p* < 0.001). Thus, our null-hypothesis was confirmed. The effect size in both cases was large (d = −2.6068, r = −0.7934 and d = −6.6177, r = −0.9514 for A1 and A2, respectively). In the 2010–2017 period, roadkill locations were still closer than random points (1.84 ± 0.20 vs 1.88 ± 0.19 km in A1 and 1.38 ± 0.12 vs 1.46 ± 0.12 km in A2), albeit insignificantly (z = 1.94, *p* = 0.052 and z = 0.35, *p* = 0.72). Effect size was small in A1 (d = −0.2050, r = −0.102) and medium in A2 (d = −0.5036. r = −0.2441).

The distribution of differences of observed roadkill values in roe deer in 2002–2017 (Figure 7a) show a much higher percent of exact matches of roadkills to the model (χ^2^ = 36.5, df = 11, *p* < 0.001; d = 0.4477, r = 0.2184, ES small). The right wing of the distribution with the biggest differences is represented by the extreme end of the highway A1, >294 km, which was excluded by the pathway model as a built-up area.

In wild boar and red deer, there were no differences in distance between the model-predicted pathways and the roadkill locations compared to the distance between pathways and random locations (all Wilcoxon test results not significant, alternative hypothesis confirmed). However, the distribution of difference values for wild boar (Figure 7b) was not equal, with the higher share of exact matches of the roadkill (χ^2^ = 29.5, df = 11, *p* < 0.002; d = 0.8711, r = 0.3993, ES medium to large). The right wing of distribution with the biggest differences is represented by the extreme ends of the highway A1, >294 km and the A2, >126 km, both being built-up areas.

For red deer (Figure 7c), roadkill locations and random points yielded the same differences from the model prediction, and the distribution of difference values is equal (χ^2^ = 8.1, df = 10, *p* = 0.61). Red deer roadkills in the built-up areas at the ends of highways A1 and A2 were not observed.

There were no differences in the distance between the model-predicted pathways and the roadkill clusters in roe deer, wild boar or the three species clusters, compared to the distance between pathways and random locations in both 2002–2009 and 2010–2017 (all Wilcoxon test results not significant). However, the distribution of difference values for roe deer (Figure 8a) was not equal, with a higher share of exact and close matches of the clusters (χ^2^ = 21.5, df = 11, *p* < 0.05). Roadkill clusters of roe deer have been observed at the beginning of the A1 in Vilnius and at the far ends of the highways A1 and A2, all these being built-up zones. Wild boar cluster locations (Figure 8b) were distributed randomly (χ^2^ = 4.8, df = 7, *p* = 0.68). Finally, cumulative roadkill clusters of all three ungulate species were better represented by the model (multi-species crossing zones), especially in terms of exact matches (χ^2^ = 31.3, df = 11, *p* < 0.01; Figure 8c). In all cases, the effect size was large (roe deer, d = 1.0272, r = 0.4569; wild boar, d = 1.63, r = 0.6325; multispecies, d = 1.0587, r = 0.4678).

## 4. Discussion

Applying habitat suitability and linkage models for red deer, roe deer and wild boar in Lithuania allowed us to forecast crossing locations on the highways A1 and A2 for each species, and according to pathway ranks. The observed roe deer roadkills in 2002–2009 were significantly closer to model-predicted pathways than random locations on both highways. In the 2010–2017 period, roe deer roadkill locations were still closer than the random points. In both periods, much higher percentages of exact matches of roadkills to the model were observed. For wild boar and red deer, there were no differences in the distance between the model-predicted pathways and the roadkill locations in comparison to the distance to random locations irrespective of the period. Still, the exact prediction of roadkill locations was good for wild boar. The biggest differences in distance from the predicted locations in roe deer, red deer and wild boar roadkills were at the beginning and end of both the A1 and A2; all these were excluded by the pathway model as built-up areas. The location of the roadkill clusters of roe deer and multi-species clusters were significantly better represented by the model than by random distribution, especially in terms of exact matches to the model-predicted pathways.

Though average roadkill cluster parameters (cluster length and cluster strength) for roe deer, wild boar and multispecies clusters did not change significantly between periods of 2002–2009 to 2010–2017, neither on the A1 nor on the A2, effect size was present in some cases. Multispecies cluster length and strength increased, while these of wild boar decreased, in both A1 and A2. For the roe deer, cluster length and strength increased in A1, cluster length did not change in A2, while cluster strength decreased in A2.

Congruence of the model to the real roadkill locations had practical significance for the implementation of the road safety measures. For example, from the model and maps, we concluded that within zone A1-4, three additional structures facilitating red deer cross-highway movement should be erected. The most suitable locations for such a purpose appear to be along the highway segments at 252 to 258 km (first structure), 262 to 265 km (second structure) and 277 to 278 km (third structure). The selection of an exact location within these three zones should be driven by engineering reasons, as well as by the conditions along the highway that are below the resolution of our simulations. Another example is the recommendation for the placement of a new under or overpass at 115–116 km of the highway A2, capable of facilitating cross-highway red deer movement (see Appendix Ab).

The usefulness of the NDVI in animal ecology, as a proxy of net primary production, is beyond doubt [31,42,43]. However, in recent years, the NDVI has been used in connection with animal distribution, abundance and dynamics [44,45,46,47]. In particular, the NDVI and other satellite-derived data are being used in predicting animal movement [45,48]. For terrestrial animals these predictions are really successful [49]. In 2009, our model was one of the first that involved predicting road crossings of the three ungulate species simultaneously [50,51]. Much later, it was shown that the NDVI is one of the main factors of landscape use and movements in wild boar [52], roe deer [53] and mule deer [54].

In Spain, wild boar and roe deer roadkills were dependent on road index (including both road density and transport intensity), slope and crops, while red deer roadkills were dependent only on road index and slope [55]. The abundance of these three species and road maintenance (fenced or not fenced) were the other significant factors for these species [56]. Increased animal abundance and traffic intensity were confirmed as resulting in increasing roadkill numbers for many other species, including ungulates, and in many countries [57,58,59], including Lithuania [60].

As many of these factors may change over time, this can lead to model accuracy decreasing in terms of evaluating species movement pathways and their conformity to locations of roadkills. In Lithuania, changes that occurred between 2002–2009 and 2010–2017 in general, as well as along the highways A1 and A2, are as follows: (i) changes in ungulate numbers, (ii) changes in transport intensity and (iii) changes in accident prevention measures (Table 6).

From 2009 until 2017, the abundance of red deer increased 2.17 times and roe deer by 1.3 times. The numbers of wild boar decreased from 2015 due to increased hunting trying to stop the spread of ASF. The roadkill number of red deer increased by 2.54, while that of roe deer increased by 3.02 and wild boar increased by 1.10. The length of wildlife fences increased by 5.66 and AADT increased by 1.29. Changes of animal abundance and AADT resulted in the increased number of the roadkills, while wildlife fencing aimed to decrease it and was successful for the A1 and A2. In case any of these changes are of significant extent, the model should be re-run and new ungulate road crossing pathways should be defined, thus increasing the achieved accuracy.

## 5. Error Evaluation

At the very basic level, the size of the error was determined by the resolution (scale) of the spatial data used in model development. Most of the digital spatial information utilized in the model was derived from 1:10,000 topographical maps of Lithuania, which implied a spatial (mapping) error of 5 to 10 m. The issue of human land development was addressed by applying certain buffer zones around town and village footprints and the mapping of barriers to wildlife movement along the highways A1 and A2 (fenced sections of the highways).

The quantification of habitat preference statements and buffer zones derived from scientific literature and expert opinion is inherently error-prone, as it is somewhat subjective and they did not always have access to the necessary information. However, we believe that the magnitude of such errors in the context of this study was negligible, particularly in light of the significant role that the distribution of human land development plays in shaping wildlife migration routes.

Using NDVI index from the August–September period to reflect of the year-to-year crop patterns may also be a source of insignificant error, as the crop planting is quite permanent and depend on the immutable soil types and moisture regimes in the territory around the highways.

Our best assessment was that potential errors cumulatively would not have changed the alignments of the simulated movement routes by more than 1 km. We acknowledged this by choosing a relatively large highway segment size (1 km) to map the high frequency crossing zones on the highways A1 and A2. In other words, we can state that the accuracy with which the model predicted the above-average interactions between the simulated movement routes and the highways is no better than 1 km.

Finally, we were not mimicking individual animals’ routes across a landscape: the models described the most efficient potential migration (long-range) routes that are based on well-established principles of habitat utilization. The generation of thousands of pathways allowed the application of statistical methods to outline and rank discrete migration corridors. Thus, the model provided a broad spatial context for examining and mitigating environmental impacts imposed by transportation corridors

## 6. Conclusions and Recommendations

The wildlife movement models for red deer, roe deer and wild boar, based on habitat suitability and linkage, represented the most probable pathways of highway crossings by these species. After testing the model validity for the modeled period 2002–2009 and subsequently for 2010–2017, we conclude that:Model-predicted pathways of roe deer significantly better described animal migration and roadkill locations than random choice in both periods, and a much higher percentage of exact matches of roadkills to the model was observed.Exact predictions for roadkill locations of the wild boar were good in both periods.The biggest differences between roadkills of red deer, roe deer and wild boar were observed in the built-up parts of the highways (vicinity of cities), and were excluded in the modeling.Roadkill clusters of roe deer and multi-species clusters including all three species were properly predicted by the models, especially in terms of locations with exact fit to the predicted pathways

We recommend wildlife movement models as an addition to the toolbox for planning WVC mitigation measures. On the basis of model validity analysis, we advise the following for increasing its predicting power:To check if registered roadkills in the area occur in the built-up areas before modeling ungulate movements, and, if so, lower the barrier sensitivity for the human settlements and outlying industrial areas (see Table 2).Before fencing considerable lengths of the highways, check the multi-species pathways and use these locations as a basis when selecting locations for the artificial wildlife structures.Re-run wildlife movement models after considerable changes in populations or habitat structure in the buffer zones.

## Figures and Tables

**Figure 1 animals-10-01345-f001:**
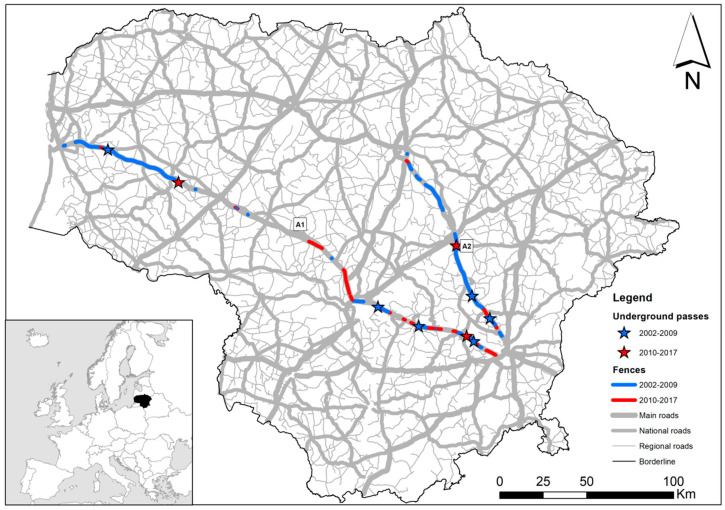
Study area: location of highways A1 and A2 in Lithuania. Also shown are wildlife highway fencing built in 2002–2009 (blue lines) and built in 2010–2017 (red lines), and wildlife underpasses (blue and red stars in 2002–2009 and 2010–2017, respectively).

**Figure 2 animals-10-01345-f002:**
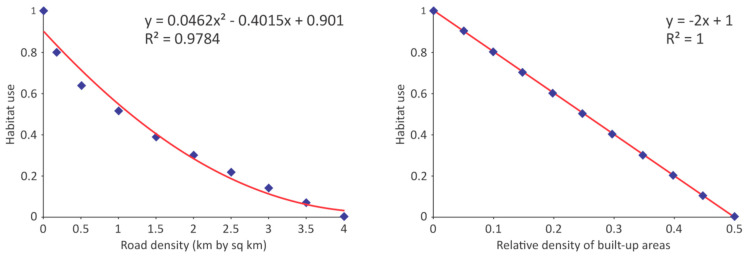
Relationship between red deer habitat utilization and: (**a**) road density, data after [30]; (**b**) relative density of built-up areas. Equations are ours.

**Figure 3 animals-10-01345-f003:**
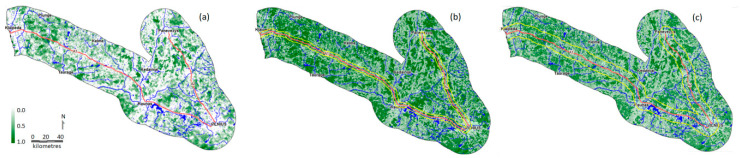
Map of modeled habitat suitability (range 0–1, where 1 is most suitable habitat) for red deer (**a**), roe deer (**b**) and wild boar (**c**) in 2007 around the highways A1 and A2 in Lithuania. Buffer size: 20 km for red deer, 3 km for roe deer and 8 km for wild boar, both shown by yellow lines. Note that the broad distribution of relatively good habitat for roe deer corresponds well with the ubiquitous nature and high adaptability of this species.

**Figure 4 animals-10-01345-f004:**
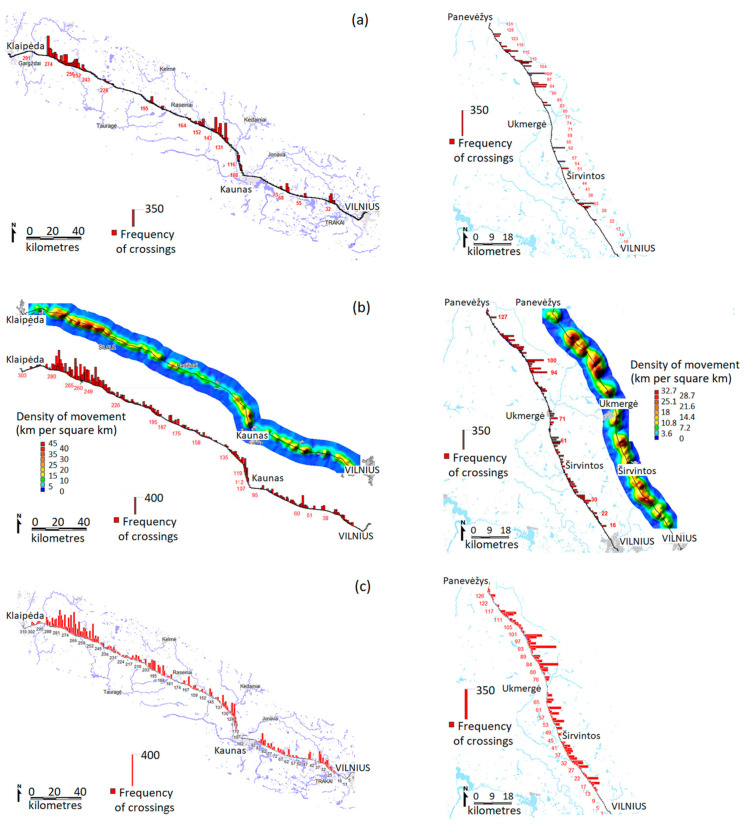
Simulated movements of the three ungulate species across highways A1 and A2 in Lithuania as of 2007: number of crossings of red deer (**a**) and wild boar (**c**) per linear km of the highway, and number of crossings and density (km per square km of the habitat) of pathways of roe deer (**b**).

**Figure 5 animals-10-01345-f005:**
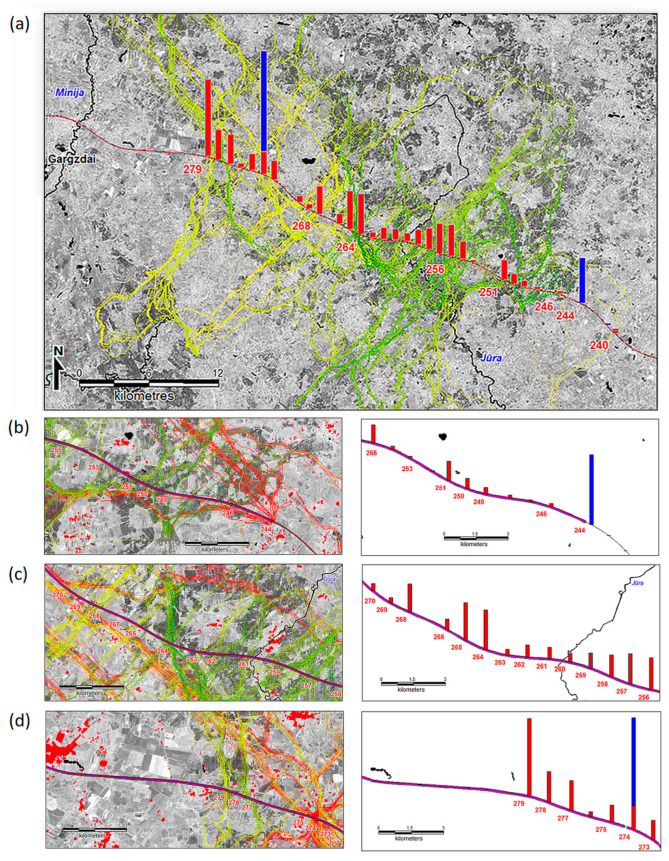
Simulated pathways and crossing frequencies of the red deer in the fenced section of highway A1, showing movement pathways across 240–280 km section with no fencing (**a**), spillover effect at the end of fence (**b**), wildlife fence paralleling red deer pathways along the highway (**c**) and concentrated pathways at the wildlife underpass (**d**). Bars depict the relative frequency of simulated movement; red bars—no fencing, maximum value 1701 crossings per linear km per year, blue bars—fenced scenario, maximum value 344 crossings per linear km per year. Scale ratio of the blue to red bars is 2.6 to 1. Yellow to green lines depict the ranking (maximum shown = 0.86) of the simulated routes (no fencing scenario), pink lines—the simulated routes of the fenced scenario.

**Figure 6 animals-10-01345-f006:**
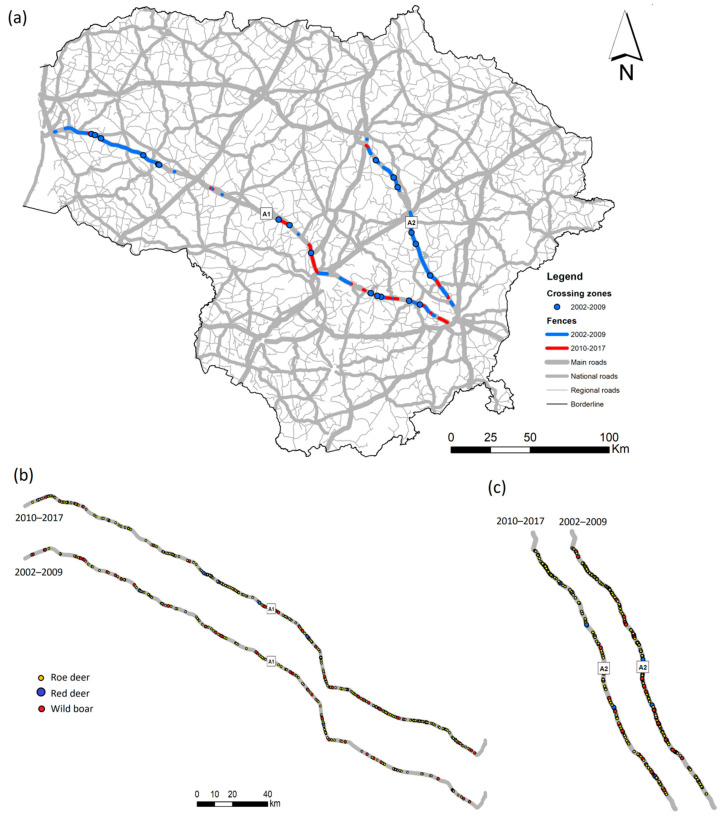
Location of multi-species crossing zones in 2002–2009 (**a**), ungulate-vehicle collision sites of highways A1 (**b**) and A2 (**c**) in 2002–2009 and 2010–2017.

**Figure 7 animals-10-01345-f007:**
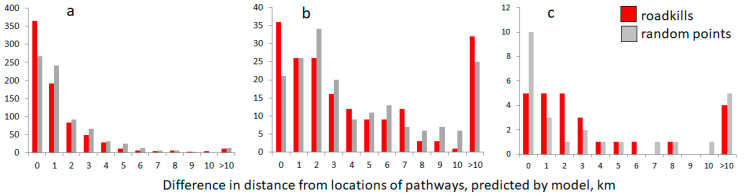
Distribution of distances between model-predicted pathways and the roadkill locations compared to distances between pathways and random locations for roe deer (**a**), wild boar (**b**) and red deer (**c**).

**Figure 8 animals-10-01345-f008:**
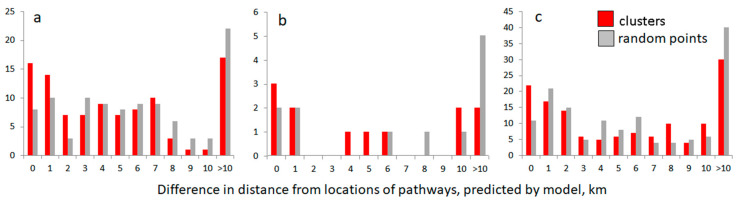
Distribution of distances between model-predicted pathways and the roadkill cluster locations compared to the distances between pathways and random locations for roe deer (**a**), wild boar (**b**) and the three species of ungulates (**c**).

**Table 1 animals-10-01345-t001:** Relative ranking of red deer, roe deer and wild boar habitat components based on a literature review and expert opinion (see also Appendix A).

Feature	Comment	Red Deer	Roe Deer	Wild Boar
Roads and Railways	Except for forest and local field roads	0.00	0.00	0.00
Built-up areas	As barriers (various buffers)	0.00	0.00	0.00
Other barriers (mines, pits, etc.)	As barriers	0.00	0.00	0.00
Lakes and rivers (major)	Obstacle but not barrier	0.05	0.01	0.01
Marshes	Obstacle but not barrier	0.05	0.05	0.05
Agriculture (fields)	>400 m from forest edge	0.10	0.40	0.30
Agriculture (fields)	0–400 m from forest edge	0.15	0.60	0.35
Grasslands/pasture	>400 m from forest edge	0.40	0.65	0.50
Berry/fruit tree plantations		0.40	0.50	0.45
Grasslands/pasture	0–400 m from forest edge	0.45	0.70	0.55
Agriculture with significant natural areas	>400 m from forest edge	0.45	0.65	0.40
Agriculture(fields)	0–250 m from forest edge	0.50	0.65	0.50
Agriculture with significant natural areas	0–400 m from forest edge	0.55	0.70 ^a^	0.55
Agriculture with significant natural areas	0–250 m from forest edge	0.60	0.75	0.60
Transitional wood/shrub lands		0.60	0.80	0.50
Coniferous forests <1250 ha		0.65	0.90	0.55
Deciduous forests <1250 ha		0.70	1.00	0.95
Riparian vegetation	Defined as >average NDVI * (50 m buffer around rivers and streams)	0.70	0.80	0.60
Riparian vegetation	Within coniferous forests > 1250 ha			0.65 ^b^
Mixed forests <1250 ha		0.75	1.00	0.80
Grasslands/pasture	0–250 m from forest edge	0.80	0.75	0.60
Coniferous forests >1250 ha		0.85	0.90	0.60
Peat bogs		0.95	0.95	0.70
Deciduous forests >1250 ha		0.95	1.00	1.00
Mixed forests >1250 ha		1.00	1.00	0.80

^a^ including urban green spaces; ^b^ used for the wild boar only; * Normalized Difference Vegetation Index.

**Table 2 animals-10-01345-t002:** Land cover types considered as barriers (B) to wildlife movement and rules of buffering the footprint of human land use to express wildlife avoidance of anthropogenic structures. The buffering rules were based on expert opinion obtained through a questionnaire distributed among wildlife biologists and forest service experts familiar with the investigated species.

Species	Small Towns and Villages	Large Towns	Cemeteries	Unvegetated Wasteland	Outlying Industrial	Quarries/Pits
Red deer	250 m	1000 m	B	B	B	B
Roe deer	footprint	100 m			B	B
Wild boar	100 m	2000 m		B	B	B

**Table 3 animals-10-01345-t003:** Eigenvectors calculated based on Saaty’s analysis for red deer (consistency ratio = 0.03), roe deer and wild boar (consistency ratio = 0.025) and eigenvectors for all three species.

Attribute	Red Deer	Roe Deer, Wild Boar	All Species
Number of crossed roads	0.0247	0.0247	
Habitat quality	0.2233	0.2644	0.18
Distance to built-up areas	0.1555	0.155	0.32
Distance to hiding cover	0.12	0.12	0.50
Proportion of the path within hiding cover	0.3121	0.27	
Path length ratio to minimum distance between entry and exit points	0.0654	0.0654	
Path length ratio to distance between entry and exit locations	0.1	0.1	

**Table 4 animals-10-01345-t004:** Basic characteristics of the four movement zones of highway A1 and the two movement zones of highway A2 according the model: KM denotes position of the zone from the highway origin in Vilnius city center in km; F—presence of fences in the zone, as for 2007; ACF—average crossing frequency per km of the highway according the model; PSM—percent of all simulated movement; APR—average pathway rank ± SD.

Zone	KM	F	Red Deer			Roe Deer			Wild Boar		
			ACF	PSM	APR	ACF	PSM	APR	ACF	PSM	APR
A1-1	1–100	Y	33	14.3	0.62 ± 0.03	47	18.3	0.54 ± 0.14	44	19.9	0.53 ± 0.10
A1-2	100–170	N	6	32.2	0.60 ± 0.04	62	12.8	0.56 ± 0.09	27	20.5	0.39 ± 0.24
A1-3	170–230	N	22	7.1	0.53 ± 0.03	34	21.1	0.44 ± 0.09	32	18.8	0.35 ± 0.16
A1-4	230–311	Y	8	46.4	0.72 ± 0.04	349	47.8	0.71 ± 0.14	67	40.7	0.62 ± 0.14
A2-1	1–74	Y	69	29.0	0.57 ± 0.03	65	47.5	0.56 ± 0.09	8	37.0	0.48 ± 0.10
A2-2	74–130	Y	53	71.0	0.71 ± 0.08	49	52.5	0.70 ± 0.10	65	63.0	0.60 ± 0.17

**Table 5 animals-10-01345-t005:** Basic characteristics of ungulate-vehicle collision clusters on highways A1 and A2 in 2002–2009 and 2010–2017. N—number of species-related accidents, n—number of accident clusters, length and strength presented as average ± SD, 95% confidence interval given in parentheses.

Period	Highway	Species	N	Species-Related Clusters		
				n	Length	Strength
2002–2009	A1	Red deer	2	-	-	-
		Roe deer	136	10	126.0 ± 13.7 (116.2–135.8)	0.364 ± 0.127 (0.273–0.455)
		Wild boar	49	3	144.3 ± 21.6 (90.7–198.0)	0.526 ± 0.111 (0.251–0.801)
		All three	187	14	129.5 ± 16.4 (120.0–139.0)	0.388 ± 0.142 (0.307–0.470)
2002–2009	A2	Red deer	8	-	-	-
		Roe deer	187	28	147.5 ± 35.8 (133.6–161.4)	0.453 ± 0.090 (0.418–0.488)
		Wild boar	50	4	138.7 ± 46.1 (65.3–212.1)	0.422 ± 0.081 (0.294–0.551)
		All three	245	42	163.6 ± 64.0 (143.6–183.5)	0.396 ± 0.102 (0.365–0.428)
2010–2017	A1	Red deer	10	-	-	-
		Roe deer	286	39	134.5 ± 23.3 (126.9–142.0)	0.445 ± 0.120 (0.407–0.484)
		Wild boar	61	4	132.8 ± 26.8 (90.2–175.3)	0.482 ± 0.141 (0.258–0.706)
		All three	357	57	141.9 ± 35.0 (132.6–151.2)	0.464 ± 0.129 (0.430–0.500)
2010–2017	A2	Red deer	6	-	-	-
		Roe deer	156	26	146.6 ± 43.1 (129.1–164.0)	0.431 ± 0.111 (0.386–0.476)
		Wild boar	25	1	101.0	0.334
		All three	187	30	153.0 ± 46.6 (135.6–170.4)	0.445 ± 0.117 (0.401–0.489)

**Table 6 animals-10-01345-t006:** Changes in numbers and roadkills of ungulates, annual average daily traffic intensity on the main roads (AADT) and length of wildlife fences (L) in Lithuania in 2002–2017.

Year	Abundance According National Survey ^1^			Number of Roadkilled Animals ^2^			L, km ^3^	AADT ^3^
	Red Deer	Roe Deer	Wild Boar	Red Deer	Roe Deer	Wild Boar		
2002	11,098	69,276	24,050	5	150	23	81	5035
2009	18,978	112,091	50,126	13	527	101	142	7278
2017	41,266	143,433	19,141	33	1591	111	804	9413

Data sources: ^1^ Ministry of the Environment of the Republic of Lithuania, ^2^ Nature Research Centre and authors data, ^3^ Lithuanian Road Administration under the Ministry of Transport and Communications.

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
