# Peer review of "Habitat Suitability Based Models for Ungulate Roadkill Prognosis"

_animals, 2020, doi:10.3390/ani10081345_

Round 1

Reviewer 1 Report

In this paper the authors modelled red deer, roe deer and wild boar movements and crossings of the two highways 11 in Lithuania.

I found the topic and the aims of the paper quite interesting. However there are a lot of issues in the methods used and poor bibliographic support that make the paper difficult to read.

The title should be more focused on the case study. It is not clear why the NDVI is mentioned in the title when it is not the main focus of the paper.

Row 86-89. Buffers are chosen but without a bibliographic reference.

Row 101. The Habitat suitability ranking is said to be based on the bibliography but there are no references.

The rankings in Table 1 seem to be very subjective.

Row 111: In Assessment of the human impacts 3 methods are used for the three species without specifying the reason and in any case there does not seem to be a strong bibliographic base.

Riow 135. It is not clear why only the August / September period is used for the calculation of the NDVI, among other things the least representative for the three species.

Row 149. I don't understand the genesis of the formulas used.

Row 173. The choice of barriers is subjective.

Row 207. Where do these buffers come from?

Row 216. Why a km?

Model testing introduces additional methods that are not adequately explained.

The results seem to be very interesting, but the methods from which they derive are too cumbersome, therefore the doubt remains that we have many inaccuracies.

For these reasons, I suggest asking for a major change to the methods. In fact, it is necessary to simplify the metology and seek greater support from the bibliography.

Author Response

Dear Reviewer, thank you for comments, please find answers in the attached file.

Reviewer 2 Report

Comments to the Author

Research on wildlife-vehicle collisions helps ecologists to understand causes and patterns of animal road-kills. In the best cases findings aim to modify road attractiveness, to reduce the risk of animal’s collision with vehicles and to make a contribution to animal conservation.

Authors of the manuscript “NDVI based habitat models for ungulate roadkill prognosis” aimed to test the efficacy of using NDVI based movement simulations of red deer, roe deer and wild boar to predict crossing locations for these species at the two highways in Lithuania. The simulations were modeled using regional habitat suitability and linkage models and calculated crossing zones in 2009. These locations were compared with wildlife-vehicle collision data of 2002-2009 and 2010-2017. Observed collision hotspots (for roe deer and wild boar), identified using KDE+, and locations (red deer) as well as random points were compared to the predicted pathways of 2009 to evaluate the prognostic capability of the model. Giving the performance of the simulation models to forecast crossing locations and pathways of red deer, roe deer and wild boar these models provide a modern approach to predict hotspots for wildlife-vehicle collisions and to improve the implementation of mitigation measures.

Overall, I found that the study at hand provides some interesting research and that the manuscript is well written. However, in reviewing the manuscript I found a few inconsistencies and parts of the paper that should be improved before publishing. For example, here and there I have difficulties in understanding how some numbers have been derived. As I have several criticisms that should be addressed before polishing the style of writing I do not provide any editorial suggestions/corrections, but I suggest authors to check the whole manuscript for edits and some possible technical improvements before resubmission.

Below please find my suggestions/criticism, which I hope to improve your manuscript.

Simple summary

L 19               Please introduce the abbreviation first

L 19               It is just fine to use WVA, but in literature WVC (and its plural WVCs) is used more often. Maybe your paper would be found more often when using WVC instead?

Introduction

L 66             If it is Wildlife Simulation Models, isn’t it WSM instead of WMS then?

L 67 ff           WMS models –> models is already included in the abbreviation. Would WMS models then be Wildlife simulation models models (or Wildlife models simulation models regarding my point above)?

Materials and methods

L 89 ff              I missed information why you used different buffer for red deer, roe deer and wild boar.

L 93 f               how long are the fences on the 311 km (A1) and 136 km (A2) in 2002-2009 and 2010-2017?

L 100 ff            I have not quite understood how the information of Table S1 provides the numbers in Table 1. I would be very grateful if you could explain this a bit more, at least to me, maybe using an example?

L101 & 104     Do you mean Table S1 instead?

L 107              Can you provide a reference for this statement?

L 149             Although already explained a bit above, I don’t understand why red deer does not have NDVI and roe deer and wild boar don't have the Rdf in the equations. It can just be me and I would be very happy if you could explain this shortly to me. Thank you.

L 152              small r in roe deer

L 207 f             m or meters? Please be consistent

L 217 f             where is the median in Table 4?

L 232 f        311-290 =21 km; 136-128= 8 km. Why are the buffer different?

L 235 f       Please check your null-hypothesis. e.g. Anderson, D. R., Burnham, K. P., & Thompson, W. L. (2000). Null hypothesis testing: problems, prevalence, and an alternative. The journal of wildlife management, 912-923.

L 238 f             You have already introduced the abbreviations in L226ff

L246                What is MVC?

L 249               which program/version did you use?

Results

L 253               what does s mean?

L 262               Figure 3: Maybe you could illustrate the above mentioned species-specific buffers zones here as well

L 272               Table 4: Does KM stand for kilometer? Where are the medians mentioned in the text?

L 281               I have never seen boar on its own. I’d rather include wild (boar) instead

L 308 ff            This is an interesting paragraph and I would rather include it in the Discussion.

L 317               Figure 4: km per or km by – text below vs. text insight the figure

L 320               Figure 5: Please include what a-d means (section A1-1 to A1-4);

344 crossings… in which time?

L 335 ff            This would be good in the Discussion

L 340               258 for both together? What were the numbers each?

L 356 ff            I'd recommend to include information about effect size and confidence interval instead of p values. e.g. Nakagawa, S., & Cuthill, I. C. (2007). Effect size, confidence interval and statistical significance: a practical guide for biologists. Biological reviews, 82(4), 591-605.

L 381               I don’t find the information above that you also clustered for all together. Could you let me know where it is written?

Discussion

L 401               I find the discussion itself quite short. E.g. could you discuss the special role/result of red deer a bit further?

L 430 ff            are these the annual hunting bag data? In general it is quite difficult to estimate wildlife density and variations in hunting bag data can have different reasons, as you know. If the collision numbers increase, it is likely that it indicates an increase in wildlife density as well (e.g. Hothorn et al. 2015)

Supplement

Table S1                      How did you put this information into numbers? Already commented above

Table S3                      bold (text)

I wish authors good luck in resubmission!

Author Response

(The authors gave the same response as above.)

Round 2

Reviewer 1 Report

The text has been implemented sufficiently.